# A Practical Guide to Full Value of Vaccine Assessments

**DOI:** 10.3390/vaccines12020201

**Published:** 2024-02-16

**Authors:** Caroline Trotter, Birgitte Giersing, Ann Lindstrand, Naor Bar-Zeev, Tania Cernuschi, Lauren Franzel-Sassanpour, Martin Friede, Joachim Hombach, Maarten Jansen, Mateusz Hasso-Agopsowicz, Mitsuki Koh, So Yoon Sim, Dijana Spasenoska, Karene Hoi Ting Yeung, Philipp Lambach

**Affiliations:** 1Disease Dynamics Unit, Department of Veterinary Medicine, University of Cambridge, Cambridge CB3 0ES, UK; 2Imperial College, London W12 7TA, UK; 3Immunization Department, World Health Organization, 1202 Geneva, Switzerland; giersingb@who.int (B.G.); franzell@who.int (L.F.-S.); sims@who.int (S.Y.S.); yeungh@who.int (K.H.T.Y.)

**Keywords:** full value of vaccine assessment, vaccine development, vaccine pipeline, vaccine value

## Abstract

Articulating the wide range of health, social and economic benefits that vaccines offer may help to overcome obstacles in the vaccine development pipeline. A framework to guide the assessment and communication of the value of a vaccine—the Full Value of Vaccine Assessment (FVVA)—has been developed by the WHO. The FVVA framework offers a holistic assessment of the value of vaccines, providing a synthesis of evidence to inform the public health need of a vaccine, describing the supply and demand aspects, its market and its impact from a health, financial and economic perspective. This paper provides a practical guide to how FVVAs are developed and used to support investment in vaccines, ultimately leading to sustained implementation in countries. The FVVA includes a range of elements that can be broadly categorised as synthesis, vaccine development narrative and defining vaccine impact and value. Depending on the features of the disease/vaccine in question, different elements may be emphasised; however, a standardised set of elements is recommended for each FVVA. The FVVA should be developed by an expert group who represent a range of stakeholders, perspectives and geographies and ensure a fair, coherent and evidence-based assessment of vaccine value.

## 1. Introduction

Vaccination is the most successful public health intervention after clean water. The World Health Organisation (WHO) estimates that over 51 million deaths are expected to be averted due to vaccines against 14 pathogens administered over the period from 2021 to 2030 (on average 5.2 million per year) [1]. There has been an expansion in the number of vaccines in the routine childhood immunisation programme, together with new vaccines being recommended for administration to pregnant women, teenagers and adults [2]. However, there are still barriers in developing and implementing new vaccines to achieve a public health impact [3]. This is particularly the case for diseases with the highest burden in low- and middle-income countries (LMICs). The WHO promotes the development of vaccines where there is the highest unmet public health need and greatest potential for impact. It does so in a variety of ways, including, for example, by publishing technical roadmaps and preferred product characteristics (PPCs, which specify product attributes and their preferential characteristics [4]) for vaccines. It has also been recognised that articulating the wide range of health, social and economic benefits that vaccines offer may help to overcome obstacles such as resource needs in the vaccine pipeline, from clinical development to bottlenecks to use at a public health scale, e.g., the prioritization of interventions by governments and funders [5,6].

A framework to guide the assessment and communication of the value of a vaccine—the Full Value of Vaccine Assessment (FVVA)—has been developed by the WHO [7]. This framework is intended to provide greater consistency than previous vaccine investment cases [8]. In addition, a number of experts have commented on the approach developed, providing views on the theoretical underpinnings and rationale for an FVVA [9].

The FVVA framework offers a holistic assessment of the value of vaccines, providing a synthesis of evidence to inform the public health need of a vaccine, describing the supply and demand aspects, its market and its impact from a health, financial and economic perspective. The ultimate goals of an FVVA are to accelerate the development of vaccines that meet a country’s needs and preferences, supporting the evaluation of vaccines and ultimately sustained introduction in countries. Lead by an independent technical agency (usually the WHO), the process of developing an FVVA brings together relevant national, regional and global experts in a working group, establishing lines of communication and alignment among these to gather, evaluate and synthesize evidence on the value of vaccines from a range of perspectives. Key stakeholders and audiences for the FVVA include the vaccine research and development (R&D) community; funders of research and vaccine implementation; vaccine market experts; global policy makers; regulatory authorities, national policy makers and programme managers; immunisation partner organisations; and civil society organisations. An FVVA should be considered a critical analysis to inform prioritization for investment and the eventual uptake of vaccines. The latter has gained increasing relevance, as the FVVA framework helps inform country decision making [10,11] by providing countries with estimates of the full value a vaccine can bring and thus empowers their decision making. This is particularly important in a context where an increasing number of decisions about the selection of appropriate newly emerging vaccines need to be made [12,13]. Of note, Donadel et al. found that policy makers place increasing emphasis on economic evaluations of vaccines [14].

Over the past years, FVVAs have become increasingly established, with the Group B Streptococcus (GBS) vaccine’s FVVA [15] being the first example covering all elements of an FVVA in their entirety. Since the publication of the GBS FVVA, substantial additional funding towards vaccine development has been granted. Extensive work towards an FVVA on other vaccines, including Group A Streptococcus (GAS) [16], tuberculosis (TB) [17,18] and influenza [19], is summarised online [20].

In this paper, we provide an update to previously published early considerations on the FVVA concept and describe in further detail its practical application. In particular, we describe the elements that an FVVA should include and how it can be used as an instrument by different stakeholders. We also describe when and how an FVVA should be developed and discuss how this fits into the overall WHO support to translate R&D efforts into country implementation. We reflect on our perspectives and experiences of developing the FVVA framework from being a concept to being a practical tool.

## 2. What Should an FVVA Include?

An FVVA can be viewed as a compendium of different elements that define the full value of a vaccine from a range of different perspectives. These elements are illustrated in Figure 1. Some elements will be informed by literature reviews and stakeholder consultation, whereas other elements will require specific research studies to be commissioned. An FVVA will therefore be supported by an extensive body of literature and evidence.

### 2.1. Synthesis

The first column in Figure 1 represents elements that provide the rationale for a vaccine (i.e., high-level global public health need), methodology of the FVVA and importantly, synthesis of both the overall findings of the FVVA and key evidence and research gaps. The latter can be used to inform the ongoing research agenda, and it highlights that an FVVA can best be considered a living document that can be updated periodically rather than presenting a snapshot at only one point in time. The synthesis of the overall findings is essential for effective communication and, in particular, making the FVVA accessible to a wider range of actors. This will also facilitate a common understanding and help in stakeholder alignment, leading to action.

### 2.2. Vaccine Development Narrative

The second column of Figure 1 shows the elements that describe the vaccine characteristics and status of vaccine development. This will include a narrative assessment of the key challenges in, and potential barriers to, vaccine development as well as an up-to-date evaluation of the vaccine pipeline. These elements most closely align with the remit of the Product Development for Vaccines Advisory Committee (PDVAC [20]). To fulfil the purpose of the FVVA as a compendium, some elements, such as Preferred Product Characteristics (PPCs, which describe the WHO’s preferences for vaccine parameters—principally, indications for vaccination, target groups, immunization strategies and the clinical data required for the assessment of safety and efficacy), will be restated here. Usually, the PPCs will inform other elements of the FVVA—for example, as input parameters in the modelling of a vaccine’s impact where future vaccine characteristics are as yet unknown. For TB vaccines, the FVVA actually assessed two different PPCs—one for adults and adolescents and one for infants. It is also in this section where the size of the required investment to develop a vaccine would be estimated and discussed. Additional considerations could include the cost of goods, price of a vaccine, or potential financing mechanisms such as through Gavi, the PAHO’s Revolving Fund or other means. An important barrier to vaccine development from the perspective of the manufacturer is uncertainty around their return on investment. To be inclusive of vaccine developer stakeholders, it is desirable to also include financial and global demand analysis, as was the case for the GBS FVVA.

### 2.3. Defining Vaccine Impact and Value

The third column of Figure 1 includes the main elements where most new research is likely required to inform the FVVA and therefore represents the main novel substance of the compendium. These elements most closely align with the remit of the Immunization and Vaccine-Related Implementation Research Advisory Committee (IVIR-AC) [21]. Ultimately an FVVA serves to inform global and national policy-making bodies such as the Strategic Advisory Group of Experts on Immunization (SAGE) or NITAGs [22]. For these downstream uses, the FVVA may need to be updated from the original if there is a considerable lag between FVVA publication and phase 3 results.

For these elements, there is some flexibility in the range of analyses presented, and their emphasis may vary according to the properties of the vaccines and epidemiology of disease. For example, considering the transmission dynamics of some infections is crucially important for understanding vaccine impact, whereas this is less the case for other infections, such as GBS, where the vaccination of pregnant women is unlikely to influence the dynamics of GBS, an organism that is widely carried in the population. Likewise, assessing the impact of vaccination on antimicrobial resistance (AMR) may be a critical factor for some infections where there are high levels of resistance or antibiotic use but less so where antibiotics are not used or where pathogens remain susceptible. The modelling of disease burden and vaccine impact in terms of DALYs, cases, deaths averted and cost-effectiveness is now well appreciated as a methodology to inform vaccine decision making. Such analyses are an essential component of the FVVA framework and would be expected to conform to high standards, demonstrated, for example, through the use of internationally recognised guidelines and checklists, such as GATHER [23], CHEERS [24] and the ISPOR Modelling Good Research Practices [25]. In particular, the uncertainty of outcomes should be appropriately represented.

There is clearly opportunity for both choice and innovation in assessing the value of a vaccine based on information needs identified by stakeholders. The FVVA includes evidence-based assessment methods driven by policy questions and decision contexts, including, among others, vaccine impact modelling; the cost of disease burden, development and delivery; investment impact; and cost-effectiveness [7]. More practically, within an FVVA, the choice of analysis should be well justified, particularly where methods that would be less familiar than a standard cost-effectiveness analysis are used. In the case of the FVVA for GBS vaccines [15], a cost-effectiveness study was performed from the perspective of the health-care provider, with results presented in terms of net monetary benefit. Sensitivity analyses were presented on some of the normative assumptions that may vary between different decision makers, such as the QALY loss assigned to a stillbirth [26]. For GAS vaccines, the global societal gains from prospective vaccines was estimated through a value-per-statistical-life approach [16]. In the case of TB vaccines, a wider range of economic analyses were performed, including analyses of the potential impact of novel vaccines on both economic growth in LMICs [17] and on health equity and financial protection in LMICs [27]. Such analyses are well justified given the epidemiology and burden of TB, a common disease affecting adults of working age. TB is also interesting in this regard because of the assessment of two different vaccine approaches, which reshaped the TB field to focus on vaccines for adults and adolescents rather than infants because of the greater potential for impact. For shigella vaccines, the association between shigella and linear growth faltering has suggested that the impact of a vaccine on child development and future productivity should be included [28]. One could also consider analyses that are pertinent to a group of pathogens—for example, those with high levels of antimicrobial resistance or antibiotic use [29]. Health security may be another dimension that requires consideration for some diseases/vaccines.

Flexibility in determining analyses presented in the FVVA presents opportunities for estimating the broader benefits of vaccination. However, there is also a need for standardisation to ensure consistency across FVVAs (or other analyses to be used by stakeholders) and enable fair comparisons to be made with other vaccines or health interventions. This will also mitigate the loss of objectivity. As Hutubessy et al. emphasise, “*the remit of an FVVA should always be aligned with the standard reference cases so as to avoid the appearance of ‘special pleading’ for particular vaccines and to avoid explicit or hidden donor-driven agendas that are not aligned with country needs*” [9]. This challenge could be addressed and balanced against the desire for innovation by having, for example, at least an economic analysis from the health provider perspective with other analyses, such as those assessing broader societal value—presented separately, as was the case for TB.

## 3. When Should an FVVA Be Developed?

There is no strict rule about when an FVVA should be developed and published. In principle, this can occur at any time in the vaccine development-to-implementation process to support discussions on vaccine prioritization; to ascertain research gaps; to generate new estimates of potential vaccine impact; or confirm the value of an intervention [30], noting that the main role of an FVVA is to encourage investment in a vaccine. To be the most useful, there should be sufficient knowledge that a vaccine is likely to be technically feasible and a degree of confidence about the likely vaccine characteristics. Thus, an FVVA is likely to come after the publication of the WHO’s PPCs. However, in some cases, an FVVA could be used to examine and optimise vaccine characteristics in advance of PPCs or used to inform a further iteration of PPCs, as for influenza. On the other hand, a vaccine should not be so sufficiently close to licensure that the FVVA does not add value. For GBS, the FVVA was published when there were two promising vaccine candidates progressing to phase II trials. For GAS, the timing was earlier, with more opportunities to influence clinical development pathways. As suggested above, an FVVA should be considered a dynamic rather than static document that can be updated and revised as further evidence becomes available and vaccine development progresses. Some have even suggested that this could be a framework to facilitate the annual monitoring of progress on completeness [6].

## 4. How Is an FVVA Developed?

For each FVVA, an expert advisory group, which should include representation from different stakeholders and experts in complementary fields, is convened. The process of developing an FVVA, including convening the expert advisory group, is led by an independent Secretariat, which ideally should be based in an independent technical agency, usually the WHO. The expert group should help to ensure that an FVVA is high quality but also that it represents a range of perspectives, i.e., this should be an inclusive process. Experts who may be included are disease-specific specialists, vaccine manufacturers, vaccine implementers, civil society actors, public health experts, epidemiologists and economists, representing all affected regions. The group should be free from conflicts of interest, as it is important that an FVVA provides a fair assessment.

## 5. How Can FVVAs Be Used as an Instrument by Stakeholders?

The findings from an FVVA can be used at various points, upstream and downstream of a licensed vaccine becoming available. Upstream, an FVVA can be used for accelerating development and incentivising investment by both vaccine manufacturers and funders of vaccine R&D. Downstream, an FVVA can be used as an evidence base to inform decisions about future vaccine introduction.

To give a specific example, Figure 2 illustrates the main findings of the GBS FVVA and how this can be translated into action by stakeholders.

A more general illustrative example of how the FVVA could be used to inform existing processes is the Gavi Vaccine Investment Strategy (VIS). The standardised template of the FVVA as described above (Section 2) aligns with criteria used by Gavi in the Vaccine Investment Strategy [31] decision-making processes. In the current 2024 VIS cycle, the work towards an FVVA was highly informative for both GBS and TB vaccines, and, indeed, the models of vaccine impact created for the FVVA were easily adapted for the VIS. It is noteworthy that the FVVA is not only useful for vaccines that are in advanced stages of development (e.g., the VIS short list) but also for informing stakeholders in earlier vaccine R&D by communicating evidence and supporting further investment in vaccine development. This is the case with GAS vaccines.

When new vaccines become available for introduction, an updated FVVA can be used as part of a package of information for global and national decision makers. This further promotes national decision making that is country-owned, evidence-based and considers the full value of vaccines. Such evidence could be used for national strategic planning or for developing recommendations for introduction recommendations, using decision support tools, such as the evidence-to-recommendation process and the CAPACITI decision support tool [32]. Further downstream, the FVVA is also likely to contain evidence that could be communicated to health workers and the public to support vaccine implementation.

## 6. How Does the FVVA Fit into the WHO’s Support Overall to Translate R&D Efforts into Country Implementation?

The FVVA is an important instrument to support the vaccine development-to-implementation pathway, but of course this sits within a broader ecosystem. Figure 3 illustrates the inclusive process through which relevant stakeholders are brought together to synthesize the required evidence and how this evidence informs subsequent policy, regulatory and country decision processes. This process also enables feedback into and updating of the underlying assumptions about the vaccine (e.g., PPCs) as the vaccine development-to-implementation pathway progresses. The WHO’s advisory committees are likely to have interest in and may have input to these different processes as indicated, with PDVAC upstream, IVIR-AC intermediate and SAGE downstream. Regional and national decision makers and implementors are essential actors.

## 7. Discussion

This paper describes the elements included in a Full Value of Vaccine Assessment, how FVVAs should be developed and their intended audiences and use cases. The FVVA can be seen as a crucial resource and a compendium of evidence on the value of a vaccine from a range of perspectives. The development of an FVVA should be inclusive of a wide range of stakeholders with an international technical agency (usually the WHO) coordinating the process. With increasingly competing demands on resources, both within and outside health (including but not limited to climate change, conflict and economic crises), there is greater need for investors in vaccines to allocate resources in an informed manner and for countries to allocate resources for immunization programmes in the most effective manner. The COVID-19 pandemic brought new vaccine technologies, which prompted a re-assessment of the value of new and perhaps even existing interventions, given the opportunities that advanced technologies can provide. The WHO leads the way in producing a variety of tools for supporting vaccine development and implementation particularly relevant to LMICs. The FVVA framework is a powerful instrument, and here, we have described how this further supports the equitable introduction of new vaccines. The WHO, through its advisory bodies, serves as a normative and guidance-setting agency, linking product innovation/development (PDVAC), implementation research (IVIR-AC) and vaccine policy making (SAGE) more cohesively with the aim of accelerating vaccine introduction at a public health scale.

Further consideration is needed with respect to evaluating the effectiveness of FVVAs in fulfilling their functions of enhancing assessment, improving decision making, facilitating communication about the value of vaccines and ultimately accelerating vaccine development and implementation. It may be difficult to attribute progress specifically to an FVVA in a causal pathway; it may be argued that the publication of an FVVA reflects building momentum behind a vaccine. Research methods such as a qualitative analysis of stakeholders’ perceptions of an FVVA may be an appropriate way of investigating this over time, particularly as more FVVAs are published.

## 8. Conclusions

The FVVA framework provides a coherent and evidence-based approach to evaluating the value of vaccines. The first FVVA was published for GBS vaccines in 2021, and there is building momentum behind this framework with more FVVAs in progress for a wide range of vaccines.

## Figures and Tables

**Figure 1 vaccines-12-00201-f001:**
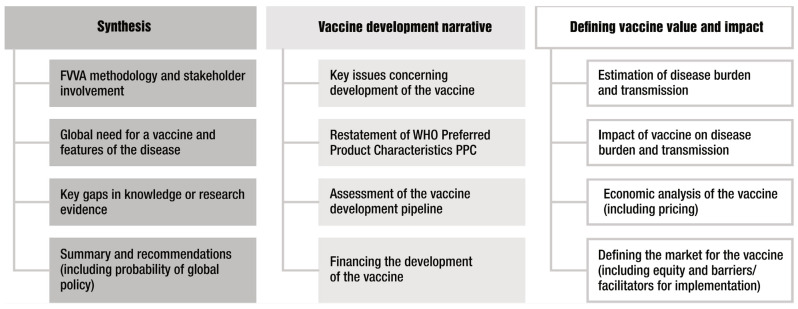
Key elements of a Full Value of Vaccine Assessment (FVVA). The columns broadly define the purpose of each element, with the third column representing areas where most new research is likely needed to inform the FVVA.

**Figure 2 vaccines-12-00201-f002:**
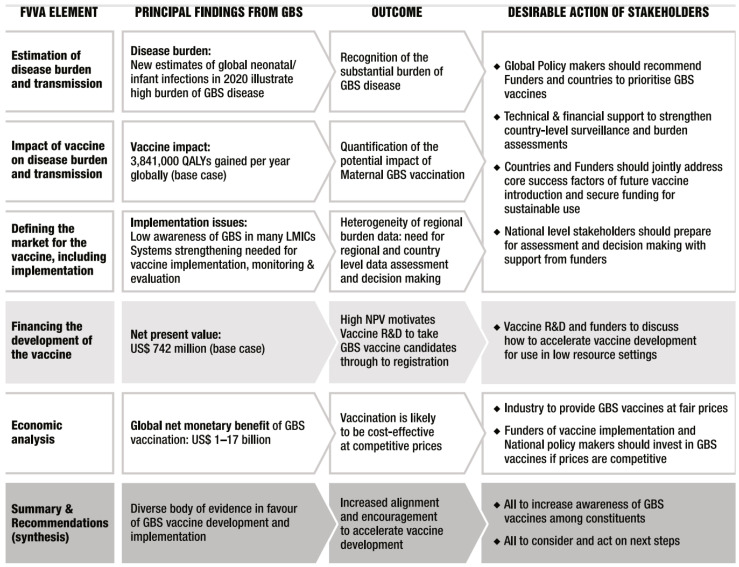
Overarching summary of the findings from the GBS FVVA and the expected impact on key stakeholders.

**Figure 3 vaccines-12-00201-f003:**
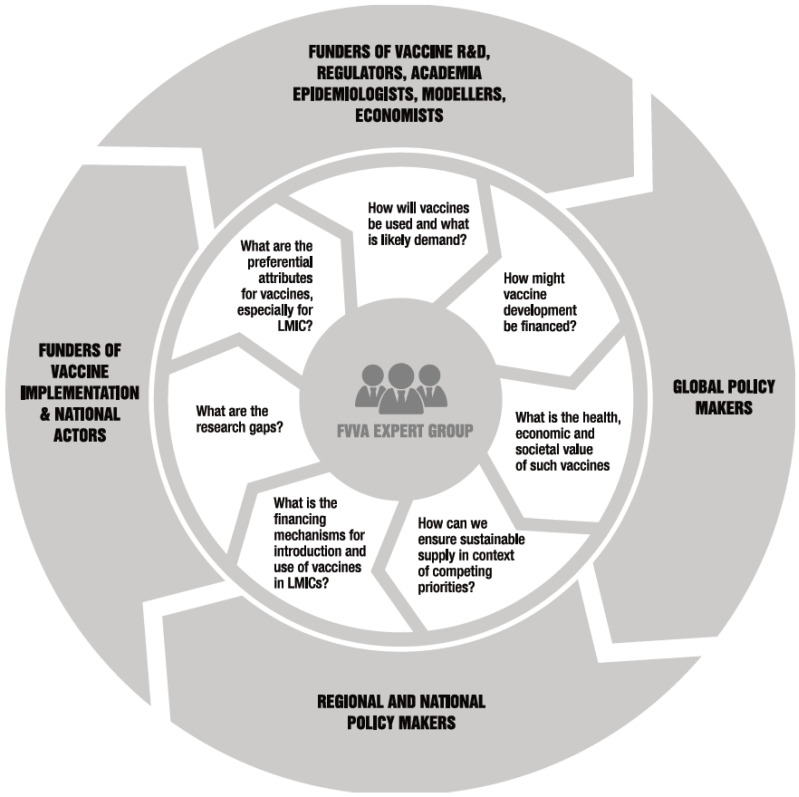
Representation of the inclusive process through which stakeholders are brought together in the vaccine development-to-implementation process.

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
