# Peer review of "A Practical Guide to Full Value of Vaccine Assessments"

_vaccines, 2024, doi:10.3390/vaccines12020201_

Round 1

Reviewer 1 Report

Comments and Suggestions for Authors

This is a perspective article aiming to serve as a practical guide regarding a novel tool for the assessment of vaccine value (FVVA), recently introduced by WHO. The article builds on the original article by Hutubessy, R. et al. (Ref no 9) and it serves as an explanation/commentary of the tool. It is clearly written and presented with appropriate figures and examples. References are also appropriate and correspond to the content of the article.

Suggestions:

The Discussion would benefit by a short paragraph on any expected barriers/difficulties in producing FVVAs. In addition, an issue which is not discussed on the Perspective is which of the stakeholders is expected to take the initiative to produce an FVVA?

Minor comments

Reference 18: Please check the link - it does not work directly from the pdf of the article

Figure 3: Bottom of graphic: Reginal instead of Regional (and National Policy Makers).

Author Response

We thank the reviewers for taking the time to evaluate our manuscript and for their helpful comments. The original comments from the reviewers are in black text, our responses to these comments are in blue text.

This is a perspective article aiming to serve as a practical guide regarding a novel tool for the assessment of vaccine value (FVVA), recently introduced by WHO. The article builds on the original article by Hutubessy, R. et al. (Ref no 9) and it serves as an explanation/commentary of the tool. It is clearly written and presented with appropriate figures and examples. References are also appropriate and correspond to the content of the article.

Suggestions:

The Discussion would benefit by a short paragraph on any expected barriers/difficulties in producing FVVAs. In addition, an issue which is not discussed on the Perspective is which of the stakeholders is expected to take the initiative to produce an FVVA?

We have reiterated in the discussion (first paragraph) that WHO will usually coordinates the FVVA process. Given the expertise within WHO (especially IVB) and their key international expert advisory groups we do not expect any barriers to either identifying vaccines for which an FVVA would be appropriate or difficulties in producing an FVVA.

Minor comments:

Reference 18: Please check the link - it does not work directly from the pdf of the article.

Apologies, this link has been replaced with the PDVAC page which includes the infographic.

Figure 3: Bottom of graphic: Reginal instead of Regional (and National Policy Makers).

This typo has been corrected.

Reviewer 2 Report

Comments and Suggestions for Authors

This manuscript is a "perspective" on the "Full Value of Vaccine Assessment" (FVVA), a framework to guide the assessment and communication of the value of a vaccine developed by WHO in 2021.

The paper aims to add some practical considerations to previoulsy published papers on the topic.

The affiliation of the Authors as well as the involvement of some of them in the development of the FVVA suggests that they are entitled to suggest a practical guide for implementing the framework. However I cannot fully understand through the paper in which way the Authors intend to suggest the guide. Is this paper a sort of summary of expert meetings? Or is it more a reflection of WHO Experts arousing from previous experiences (i.e. the GBS FVVA)? For example, Figure 1 suggests key elements of a FVVA , while Figure 2 illustrates how the main findings of the GBS FVVA could impact on stakeholders.

I think that could be usuful if the authors add a short paraghraph to clarify this aspect. In this sense, also the section "authors contribution" could be used to make clearer the contribution of each author to the paper.

Furthermore, I've just some minor remarks for the English (see below)

Comments on the Quality of English Language

P2, line 56: synthesis --> synthesize

P5 line 183-187: add colon/commas

Author Response

We thank the reviewers for taking the time to evaluate our manuscript and for their helpful comments. The original comments from the reviewers are in black text, our responses to these comments are in blue text.

This manuscript is a "perspective" on the "Full Value of Vaccine Assessment" (FVVA), a framework to guide the assessment and communication of the value of a vaccine developed by WHO in 2021.

The paper aims to add some practical considerations to previously published papers on the topic.

The affiliation of the Authors as well as the involvement of some of them in the development of the FVVA suggests that they are entitled to suggest a practical guide for implementing the framework. However I cannot fully understand through the paper in which way the Authors intend to suggest the guide. Is this paper a sort of summary of expert meetings? Or is it more a reflection of WHO Experts arousing from previous experiences (i.e. the GBS FVVA)? For example, Figure 1 suggests key elements of a FVVA , while Figure 2 illustrates how the main findings of the GBS FVVA could impact on stakeholders.

I think that could be usuful if the authors add a short paraghraph to clarify this aspect. In this sense, also the section "authors contribution" could be used to make clearer the contribution of each author to the paper.

Thank you, we have added a sentence to clarify this at the end of the discussion.

Furthermore, I've just some minor remarks for the English (see below)

Comments on the Quality of English Language

P2, line 56: synthesis --> synthesize

P5 line 183-187: add colon/commas

We have corrected these errors.

Reviewer 3 Report

Comments and Suggestions for Authors

This is an extremely well-written manuscript that is practically ready for publication now. It needs some additional details for the reader not specialised in global health matters nor vaccination policy, but I have no trouble recommending the most minor of revisions before publication.

Line 29-30: Which routine childhood immunisation programme are the authors referring to? The WHO recommended schedule or some country-specific programme?

Line 87-181: The text could do with having sub-headings corresponding to the labels on the compartments of Figure 1 so that the reader can make better sense of this section. For the non-expert, the figure is not easy to understand. The text attempts to explain the figure but without some corresponding sub-headings for each box it’s difficult to do so. Please make sure that each compartment of this figure is clearly explained. It is not necessary to do this for Figure 2.

Author Response

We thank the reviewers for taking the time to evaluate our manuscript and for their helpful comments. The original comments from the reviewers are in black text, our responses to these comments are in blue text.

This is an extremely well-written manuscript that is practically ready for publication now. It needs some additional details for the reader not specialised in global health matters nor vaccination policy, but I have no trouble recommending the most minor of revisions before publication.

Line 29-30: Which routine childhood immunisation programme are the authors referring to? The WHO recommended schedule or some country-specific programme?

We have added more context that the referenced study included vaccines against 14 pathogens.

Line 87-181: The text could do with having sub-headings corresponding to the labels on the compartments of Figure 1 so that the reader can make better sense of this section. For the non-expert, the figure is not easy to understand. The text attempts to explain the figure but without some corresponding sub-headings for each box it’s difficult to do so. Please make sure that each compartment of this figure is clearly explained. It is not necessary to do this for Figure 2.

We chose to present the subheading in the text to match the main column headings in the figure. We hope that this is clearer in the revised figure but would prefer not to add subheading for each box of the figure as we think this will make it more difficult to read.